# The economic impact of premature mortality in Cabo Verde: 2016–2020

Ngibo Mubeta Fernandes[1]*, Janilza Solange Gomes Silveira Silva[1], Domingos Veiga Varela[1], Edna Duarte Lopes[2], Janice de Jesus Xavier Soares[2]

1 National Health Observatory, National Public Health Institute, Praia, Cabo Verde, 2 Department of Research, Science and Innovation in Health, National Public Health Institute, Praia, Cabo Verde

* ngibo.fernandes@insp.gov.cv

## Abstract

Mortality analysis studies in Cabo Verde are scarce and those available are limited to short periods of analysis and to specific population groups. National mortality data reports do not quantify the burden of disease associated with premature mortality. This study estimated the years of potential life lost (YPLL), years of potential productive life lost (YPPLL) and the costs associated with them in Cabo Verde from 2016 to 2020 and aimed to determine trends of early mortality due to all causes of death. Mortality data were obtained from the Ministry of Health, Cabo Verde. Deaths that occurred from 2016 to 2020, in individuals aged between one (1) and 73 years old were analyzed by sex, age group, municipality and cause of death. YPLL, YPPLL and cost of productivity lost (CPL) were estimated using life expectancy and the human capital approach methods, respectively. There were 6100 deaths recorded in the sample population and males represented 68.1% (n = 4,154) of the reported deaths. The number of deaths verified corresponded to 145,544 YPLL, of which 69.0% (n = 100,389) were attributed to males. There were 4,634 deaths among individuals of working age, which resulted in 80 965 YPPLL, with males contributing 72.1% (n = 58,403) of the total YPPLL. The estimated CPL due to premature death was 98,659,153.23 USD. Injuries and external causes together accounted for 21,580,954.42 USD (21.9%) of CPL, while diseases of the circulatory system 18,843,260.42 USD (19.1%) and certain infectious and parasitic diseases accounted for 16,633,842.70 USD (16.9%). The study demonstrated the social and economic burden of premature mortality. The YPLL, YPPLL and CPL measures can be used to complement measures traditionally used to demonstrate the burden and loss of productivity due to premature mortality and to support resource allocation and public health decision making in Cabo Verde.

## Introduction

Mortality and average life expectancy represent simple measures to assess the health/disease process of a population. National mortality data help to characterize deaths and trends related to the causes and circumstances of deaths [1]. In addition, mortality data allow comparing

**Data Availability Statement:** All relevant data are within the paper and its Supporting Information files.

**Funding:** The authors received no specific funding for this work.

**Competing interests:** The authors have declared that no competing interests exist.

patterns within and between populations, and provide a basis for the evaluation and planning of public health programs.

Crude mortality data, however, have important limitations, such as the fact that they do not reflect the age composition of the population, do not report on preventable deaths in certain age groups and do not quantify the burden of premature mortality [2]. To fill this gap, analysts have increasingly resorted to other measures, including the years of potential life lost.

By definition, years of potential life lost (YPLL) represent an estimate of the average time a person would have lived had they not died prematurely. As an impact measure, YPLL seeks to quantify the socioeconomic burden of premature deaths [3]. Meanwhile, YPPLL quantify the average time a person would have lived if they did not die prematurely, considering the productive age of a given population. The YPPLL use the lower and upper limits of the productive life period as age cut-off and serve as a basis for calculating CPL which is the representation of YPPLL in monetary terms [4].

Cost of illness studies are considered an essential evaluation technique in the health sector. By measuring and comparing the economic burden of disease on society, these studies can help health decision-makers establish and prioritize health policies and interventions [5].

Despite significant socioeconomic and health improvements, Cabo Verde still has a high burden of mortality in the 15 to 49 age group [6]. Knowing that this age group falls within the productive life span in Cabo Verde (15 to 65 years) it is important to quantify and understand premature mortality in the archipelago.

Mortality analysis studies in Cabo Verde are scarce and those found are limited to short periods of analysis and to specific population groups [7, 8]. Moreover, a literature review on the subject did not identify studies evaluating YPLL, YPPLL or costs associated with premature mortality in the archipelago. This study sought to fill this gap, by calculating, analyzing the years of potential life lost, and cost associated with premature mortality, by municipality, sex and cause of death in Cabo Verde between 2016 and 2020.

The results of the study intend to elucidate the patterns of early mortality in the analyzed period, being able to subsidize public policies, mainly at the sanitary level. This work focused on the calculation of years of potential life lost, years of potential productive life lost and the costs associated with these in Cabo Verde from 2016 to 2020.

## Methodology

### General approach

This is an observational, cross-sectional, quantitative and analytical study using secondary data. For data analysis, all deaths that occurred in Cabo Verde from 2016 to 2020, aged between one (1) and 74 years old, were considered. Deaths that occurred in the first year of life were excluded due to the specificity of mortality in this period and due to the social and economic value that this loss represents [9, 10].

In this study mortality data were presented by cause of death according to the International Statistical Classification of Diseases and Related Health Problems 10th Revision (ICD-10) [11].

### Data sources and data collection

The study analyzed data provided by the National Directorate of Health, on an anonymized basis. Therefore, these are anonymized secondary data that, according to the legislation in force in the country, do not require authorization from the National Committee of Ethics for Health Research.

Quality control involved several aspects, at different levels. Standards of technical quality requirement for the quality of the database were ensured by technicians from the National Public Health Institute (INSP). The entire process of cleaning and structural analysis of the

data (data cleaning) was conducted, including identification of 'missing data', systematic errors in jumps and inadequate patterns of filed data.

The statistical analysis of data was performed using the SPSS statistical software (version 26) and Microsoft Excel.

## Estimation methods

Statistical analysis of data started with descriptive statistics, calculating absolute and relative frequencies of deaths that occurred in the period under analysis. Calculations of YPLL, YPPLL and CPL, and their respective absolute and relative frequencies, rates and means were determined and estimated by sex, age group, municipality and cause of death.

The Human Capital Approach was adapted, assuming that an individual would earn a constant value throughout their productive life if it were not interrupted by premature death [12].

YPLL were estimated by adapting the method proposed by Romeder and Mcwhinnie [9]. The maximum age limit was 74 years, the average life expectancy at birth for the Cape Verdean population in 2019 [13]. The YPLL calculation method for a given factor consisted of the sum of the number of deaths at each age (between 1 and 73 years old) multiplied by the remaining years of life up to 74 years of age. The number of deaths were distributed by age groups of five (5) years. In addition, the midpoint of each age groups was calculated. Subsequently, the midpoint of each age group was subtracted from 74 years, and the difference was multiplied by the number of deaths in each age group. The result represents the estimated number of years of life lost due to a specific cause of death. Thus, the following formula was used to estimate YPLL [4]:

$$YPLL = \sum_{i=1}^{73} a_i d_i = \sum_{i=1}^{73} (74 - i - 0.5)d_i \tag{1}$$

Where: $a_i$–years remaining to reach the age corresponding to the upper limit considered, when death occurs between the ages of i and i + 1 years, using the adjustment of 0.5 assuming that all deaths occurred in the middle of the year and $d_i$–number of deaths between age i and i + 1 years.

To determine the economic losses due to premature deaths during the study period, YPPLL and CPL method were used [4]:

$$YPPLL = \sum_{j=1}^{64} w_j d_j = \sum_{j=1}^{64} (65 - j - 0.5)d_j \tag{2}$$

Where: $w_j$–number of years left to complete the age corresponding to the upper limit considered when death occurs between age j and j+1 year; $d_j$–number of deaths between the age of j and j + 1 year, using the adjustment of 0.5 assuming all deaths occurred in the middle of the year; j–mean age in 5-year age groups for the productive population (15–64 years); 65 –retirement age limit in Cabo Verde [14] and 15 –minimum working age [15].

$$CPL = \sum_{j=1}^{j} YPPLL \times GDP_{Per\ capita} \tag{3}$$

Where: the Gross Domestic Product (GDP) per capita for 2017, of 3 427.49 USD (321 945 $00 *Escudos Caboverdianos*) [14] and j as defined previously.

Future values were presented in present values by applying the discount rate of 4.5% practiced by the Bank of Cabo Verde [16]. In addition, a sensitivity analysis was performed applying rates of 3% and 6% to adjust for possible fluctuations in rates [12, 17]. To determine the present value of the cost of productivity lost, the following formula was used:

$$PV = \sum_{t=0}^{T-1} CPL(1 + r)^{-t} \tag{4}$$

Where: PV–present value, CPL–future value, r–discount rate, t–mean number of years remaining to reach the age corresponding to the upper limit considered in each 5-year age group.

### Ethical considerations

As this study analyzed existing secondary data provided by the National Directorate of Health and published in Annual Health Reports, submission to the National Ethics Committee for Health Research and to the National Commission for Data Protection was not required.

## Results

### Sociodemographic characterization of subjects

**Sample characterization.**   The sample consisted of 6100 deaths, aged between 1 and 73 years and males represented 68.1% (4154) of deaths. Overall, 62.3% (3800) of deaths occurred in the age group between 35 to 64 years. The highest number of deaths occurred in the municipalities of Praia (27.7%; 1692) and São Vicente (17.6%; 1076) Table 1.

In the period under analysis, diseases of the circulatory system, certain infectious and parasitic diseases, neoplasms, diseases of the respiratory system, external causes of morbidity and mortality and injuries, poisonings and certain other consequences of external causes were responsible for 81.7% (5064) of deaths, while 18.3% (1036) of deaths were attributed to the remaining groups of causes of death. Diseases of the circulatory system were the main cause of death, accounting for 23.4% (1427) of deaths, followed by certain infectious and parasitic diseases (15.4%; 940) and neoplasms (15.3%; 935). It is worth noting that in 2020, 112 deaths recorded, were due to COVID-19 and were included the certain infectious and parasitic diseases group Table 2.

### Estimates of years of potential life lost in Cabo Verde, 2016 to 2020

From 2016 to 2020, 42.9% (6100) of deaths that occurred in individuals aged between 1 to 73 years, corresponded to 145,544 YPLL. The highest number of YPLL (21,978) were in 2020. The results obtained showed that the total mean YPLL per death in the five-year period varied from 22.6 to 25.0 Table 3.

Results showed that the leading contributors to YPLL between 2016 and 2020 were diseases of the circulatory system, certain infectious and parasitic diseases, external causes of morbidity and mortality and injuries, poisonings and certain other consequences of external causes, and accounted for 71% (103,336) of the total YPLL Table 4. The highest number of YPLL (31,509) was reported in 2020, where diseases of the circulatory system accounted for 5,913 YPLL, followed by certain infectious and parasitic diseases (5,312), external causes of morbidity and mortality (4,298), neoplasms (3,618), diseases of the respiratory system (3,287) and injuries, poisonings and certain other consequences of external causes (3,077). Combined, these groups represented 80.9% of YPLL in that year Table 4.

In the 5-year period, males accounted for 69.0% (100,389) of the total estimated YPLL and the total mean loss per death was 23.9 for both sexes, 24.2 for males and 23.2 for females. The highest proportion of YPLL for females (22.1%; 9984) was attributable to certain infectious and parasitic diseases, with an average of 27.1 YPLL per death. As for males, the largest contributors to YPLL were deaths due to external causes of morbidity and mortality (18.3%; 18394), with an average of 37 YPLL per death Table 5.

Overall, males contributed a higher number of YPLL throughout the study period (2016–2020) compared to females. The highest proportion of YPLL (62.2%; 62479) was attributed to

**Table 1. Characterization of the sample population, 2016 to 2020.**

| Variables | | Sex | | | |
|---|---|---|---|---|---|
| | | Male | | Female | |
| | | n | % | n | % |
| Age group | 1 to 4 | 74 | 1.2 | 72 | 1.2 |
| | 5 to 9 | 24 | 0.4 | 23 | 0.4 |
| | 10 to 14 | 38 | 0.6 | 24 | 0.4 |
| | 15 to 19 | 64 | 1.0 | 39 | 0.6 |
| | 20 to 24 | 151 | 2.5 | 49 | 0.8 |
| | 25 to 29 | 176 | 2.9 | 59 | 1.0 |
| | 30 to 34 | 218 | 3.6 | 78 | 1.3 |
| | 35 to 39 | 294 | 4.8 | 130 | 2.1 |
| | 40 to 44 | 348 | 5.7 | 124 | 2.0 |
| | 45 to 49 | 412 | 6.8 | 138 | 2.3 |
| | 50 to 54 | 521 | 8.5 | 193 | 3.2 |
| | 55 to 59 | 583 | 9.6 | 259 | 4.2 |
| | 60 to 64 | 500 | 8.2 | 298 | 4.9 |
| | 65 to 69 | 421 | 6.9 | 278 | 4.6 |
| | 70 to 73 | 330 | 5.4 | 182 | 3.0 |
| | **Total** | 4154 | 68.1 | 1946 | 31.9 |
| Marital status | Single | 2924 | 52.5 | 1274 | 22.9 |
| | Married / de facto union | 780 | 14 | 328 | 5.9 |
| | Divorced / Separated | 79 | 1.4 | 34 | 0.6 |
| | Widower | 37 | 0.7 | 110 | 2 |
| | Other | 2 | 0 | 1 | 0 |
| | **Total** | **3822** | **68.6** | **1747** | **31.4** |
| Municipality | Ribeira Grande | 176 | 2.9 | 61 | 1 |
| | Paul | 74 | 1.2 | 28 | 0.5 |
| | Porto Novo | 142 | 2.3 | 43 | 0.7 |
| | São Vincent | 751 | 12.3 | 325 | 5.3 |
| | Ribeira Brava | 68 | 1.1 | 20 | 0.3 |
| | Tarrafal de São Nicolau | 55 | 0.9 | 20 | 0.3 |
| | Sal | 203 | 3.3 | 105 | 1.7 |
| | Boavista | 78 | 1.3 | 35 | 0.6 |
| | Maio | 47 | 0.8 | 24 | 0.4 |
| | Tarrafal | 128 | 2.1 | 103 | 1.7 |
| | Santa Catarina | 339 | 5.6 | 187 | 3.1 |
| | Santa Cruz | 200 | 3.3 | 107 | 1.8 |
| | Praia | 1147 | 18.8 | 545 | 8.9 |
| | São Domingos | 97 | 1.6 | 57 | 0.9 |
| | São Miguel | 103 | 1.7 | 63 | 1 |
| | São Salvador do Mundo | 51 | 0.8 | 30 | 0.5 |
| | São Lourenço dos Órgãos | 52 | 0.9 | 17 | 0.3 |
| | Ribeira Grande de Santiago | 51 | 0.8 | 30 | 0.5 |
| | Mosteiros | 56 | 0.9 | 19 | 0.3 |
| | São Felipe | 169 | 2.8 | 73 | 1.2 |
| | Santa Catarina do Fogo | 34 | 0.6 | 16 | 0.3 |
| | Brava | 63 | 1 | 23 | 0.4 |
| | Others | 70 | 1.1 | 15 | 0.2 |
| | **Total** | **4154** | **68.1** | **1946** | **31.9** |

**Table 2. Number of deaths by sex and cause of death, 2016 to 2020.**

| Cause of death | Sex | | | |
|---|---|---|---|---|
| | Male | | Female | |
| | n | % | n | % |
| Diseases of the circulatory system | 890 | 14.6 | 537 | 8.8 |
| Certain infectious and parasitic diseases | 571 | 9.4 | 369 | 6 |
| Neoplasms [tumors] | 564 | 9.2 | 371 | 6.1 |
| External causes of morbidity and mortality | 497 | 8.1 | 72 | 1.2 |
| Diseases of the respiratory system | 435 | 7.1 | 191 | 3.1 |
| Injury, poisoning and certain other consequences of external causes | 414 | 6.8 | 71 | 1.2 |
| Diseases of the digestive system | 228 | 3.7 | 88 | 1.4 |
| Symptoms, signs and abnormal clinical and laboratory findings, not elsewhere classified | 150 | 2.5 | 75 | 1.2 |
| Mental and behavioral disorders | 148 | 2.4 | 13 | 0.2 |
| Endocrine, nutritional and metabolic diseases | 106 | 1.7 | 56 | 0.9 |
| Diseases of the nervous system | 83 | 1.4 | 32 | 0.5 |
| Diseases of the genitourinary system | 50 | 0.8 | 27 | 0.4 |
| Diseases of the skin and subcutaneous tissue | 6 | 0.1 | 2 | 0.03 |
| Diseases of the blood and blood-forming organs and certain disorders involving the immune mechanism | 5 | 0.1 | 9 | 0.1 |
| Diseases of the musculoskeletal system and connective tissue | 3 | 0.05 | 1 | 0.02 |
| Congenital malformations, deformations and chromosomal abnormalities | 2 | 0.03 | 2 | 0.03 |
| Certain conditions originating in the perinatal period | 2 | 0.03 | 1 | 0.02 |
| Pregnancy, childbirth and puerperium | 0 | 0 | 29 | 0.5 |
| **Total** | **4154** | **68.1** | **1946** | **31.9** |

individuals between 30 to 59 years for males. While for females, the highest proportion of YPLL corresponded to the age group 1 to 4 years (11.3%; 5112) followed by 35 to 39 years (10.5%; 4745) (Fig 1).

## Estimation of years of potential productive life lost

In the study period, 80 965 YPPLL were lost and were attributed to the 4,634 deaths that occurred in individuals the productive age group. There was a greater loss of years of productive life among men (72.1%; 58403) when compared to women (27.9%; 22563). The annual average loss was 11680.8 YPPLL for men and 3926.2 for women, respectively Table 6.

Overall, the highest YPPLL values were recorded in 2018 and 2020, and YPPLL varied from 14,043 to 17,328 Table 7. In the 5-year study period, the largest contributors to YPPLL were external causes of morbidity and mortality, responsible for 17.9% (14,525) of total YPPLL, followed by certain infectious and parasitic diseases 16.8% (13,583) and diseases of the circulatory

**Table 3. Number of deaths and years of potential life lost by sex, Cabo Verde, 2016 to 2020.**

| Year | | Male | | | | Female | | | | Total | | |
|---|---|---|---|---|---|---|---|---|---|---|---|---|
| | n | YPLL | YPLL | Mean | n | YPLL | YPLL rate | Mean | n | YPLL | YPLL rate | Mean |
| 2016 | 787 | 19920 | 78.4 | 25.3 | 379 | 9259 | 37.3 | 24.4 | 1166 | 29179 | 58.1 | 25.0 |
| 2017 | 723 | 17095 | 66.2 | 23.6 | 347 | 8552 | 34.0 | 24.6 | 1070 | 25647 | 50.4 | 24.0 |
| 2018 | 868 | 21513 | 82.1 | 24.8 | 425 | 9585 | 37.8 | 22.6 | 1293 | 31098 | 60.3 | 24.1 |
| 2019 | 818 | 19884 | 74.8 | 24.3 | 360 | 8228 | 32.1 | 22.9 | 1178 | 28111 | 53.8 | 23.9 |
| 2020 | 958 | 21978 | 81.5 | 22.9 | 435 | 9532 | 36.8 | 21.9 | 1393 | 31509 | 59.6 | 22.6 |

**Table 4. Evolution of years of potential life lost by cause of death, 2016 to 2020.**

| Cause of death | Reference year | | | | |
|---|---|---|---|---|---|
| | **2016** | **2017** | **2018** | **2019** | **2020** |
| Certain conditions originating in the perinatal period | 71 | 0 | 71 | 71 | 0 |
| External causes of morbidity and mortality | 5229 | 3911 | 4020 | 3663 | 4298 |
| Diseases of the genitourinary system | 200 | 247 | 298 | 102 | 248 |
| Diseases of the circulatory system | 4820 | 4343 | 5003 | 4724 | 5913 |
| Diseases of the digestive system | 1523 | 1478 | 1250 | 1568 | 1460 |
| Diseases of the respiratory system | 2317 | 2154 | 3869 | 2848 | 3287 |
| Mental and behavioral disorders | 827 | 896 | 859 | 518 | 920 |
| Endocrine, nutritional and metabolic diseases | 671 | 682 | 691 | 930 | 609 |
| Diseases of the nervous system | 860 | 626 | 855 | 557 | 901 |
| Pregnancy, childbirth and puerperium | 83 | 203 | 156 | 452 | 291 |
| Certain infectious and parasitic diseases | 4697 | 4284 | 5734 | 4256 | 5312 |
| Injury, poisoning and certain other consequences of external causes | 3647 | 3042 | 3483 | 3344 | 3077 |
| Congenital malformations, deformations and chromosomal abnormalities | 71 | 57 | 0 | 138 | 0 |
| Diseases of the musculoskeletal system and connective tissue | 0 | 52 | 73 | 0 | 12 |
| Diseases of the skin and subcutaneous tissue | 22 | 43 | 7 | 2 | 26 |
| Diseases of the blood and blood-forming organs and certain disorders involving the immune mechanism | 0 | 42 | 169 | 171 | 60 |
| Symptoms, signs and abnormal clinical and laboratory findings, not elsewhere classified | 833 | 760 | 1258 | 1295 | 1483 |
| Neoplasms [tumors] | 3311 | 2831 | 3306 | 3476 | 3618 |
| **Total** | **29179** | **25647** | **31098** | **28111** | **31509** |

**Table 5. Years of potential life lost by sex and cause of death, 2016 to 2020.**

| Cause of death | Male | | | Female | | | Total | | |
|---|---|---|---|---|---|---|---|---|---|
| | YPLL | % | YPLL per death | YPLL | % | YPLL per death | YPLL | % | YPLL per death |
| Certain conditions originating in the perinatal period | 142 | 0.1 | 71.0 | 71.0 | 0.2 | 71.0 | 213.0 | 0.1 | 71.0 |
| External causes of morbidity and mortality | 18393.5 | 18.3 | 37.0 | 2726.5 | 6.0 | 37.9 | 21120.0 | 14.5 | 37.1 |
| Diseases of the genitourinary system | 746.0 | 0.7 | 14.9 | 347.5 | 0.8 | 12.9 | 1093.5 | 0.8 | 14.2 |
| Diseases of the circulatory system | 15921.0 | 15.9 | 17.9 | 8881.5 | 19.7 | 16.5 | 24802.5 | 17.0 | 17.4 |
| Diseases of the digestive system | 5118.0 | 5.1 | 22.4 | 2159.0 | 4.8 | 24.5 | 7277.0 | 5.0 | 23.0 |
| Diseases of the respiratory system | 9793.0 | 9.8 | 22.5 | 4681.0 | 10.4 | 24.5 | 14474.0 | 9.9 | 23.1 |
| Mental and behavioral disorders | 3664.0 | 3.6 | 24.8 | 354.5 | 0.8 | 27.3 | 4018.5 | 2.8 | 25.0 |
| Endocrine, nutritional and metabolic diseases | 2455.5 | 2.4 | 23.2 | 1126.0 | 2.5 | 20.1 | 3581.5 | 2.5 | 22.1 |
| Diseases of the nervous system | 2603.0 | 2.6 | 31.4 | 1195.5 | 2.6 | 37.4 | 3798.5 | 2.6 | 33.0 |
| Pregnancy, childbirth and puerperium | NA* | NA* | NA* | 1183.5 | 2.6 | 40.8 | 1183.5 | 0.8 | 40.8 |
| Certain infectious and parasitic diseases | 14297.5 | 14.2 | 25.0 | 9984.0 | 22.1 | 27.1 | 24281.5 | 16.7 | 25.8 |
| Injury, poisoning and certain other consequences of external causes | 13632.0 | 13.6 | 32.9 | 2959.5 | 6.6 | 41.7 | 16591.5 | 11.4 | 34.2 |
| Congenital malformations, deformations and chromosomal abnormalities | 137.5 | 0.1 | 68.8 | 127.5 | 0.3 | 63.8 | 265.0 | 0.2 | 66.3 |
| Diseases of the musculoskeletal system and connective tissue | 84.5 | 0.1 | 28.2 | 51.5 | 0.1 | 51.5 | 136.0 | 0.1 | 34.0 |
| Diseases of the skin and subcutaneous tissue | 60.5 | 0.1 | 10.1 | 38.0 | 0.1 | 19.0 | 98.5 | 0.1 | 12.3 |
| Diseases of the blood and blood-forming organs and certain disorders involving the immune mechanism | 142.5 | 0.1 | 28.5 | 298.0 | 0.7 | 33.1 | 440.5 | 0.3 | 31.5 |
| Symptoms, signs and abnormal clinical and laboratory findings, not elsewhere classified | 3730.5 | 3.7 | 24.9 | 1897.0 | 4.2 | 25.3 | 5627.5 | 3.9 | 25.0 |
| Neoplasms [tumors] | 9468.0 | 9.4 | 16.8 | 7072.5 | 15.7 | 19.1 | 16540.5 | 11.4 | 17.7 |
| **Total** | **100389.0** | **100** | **24.2** | **45154.0** | **100** | **23.2** | **145543.0** | **100.0** | **23.9** |

* NA–Not applicable

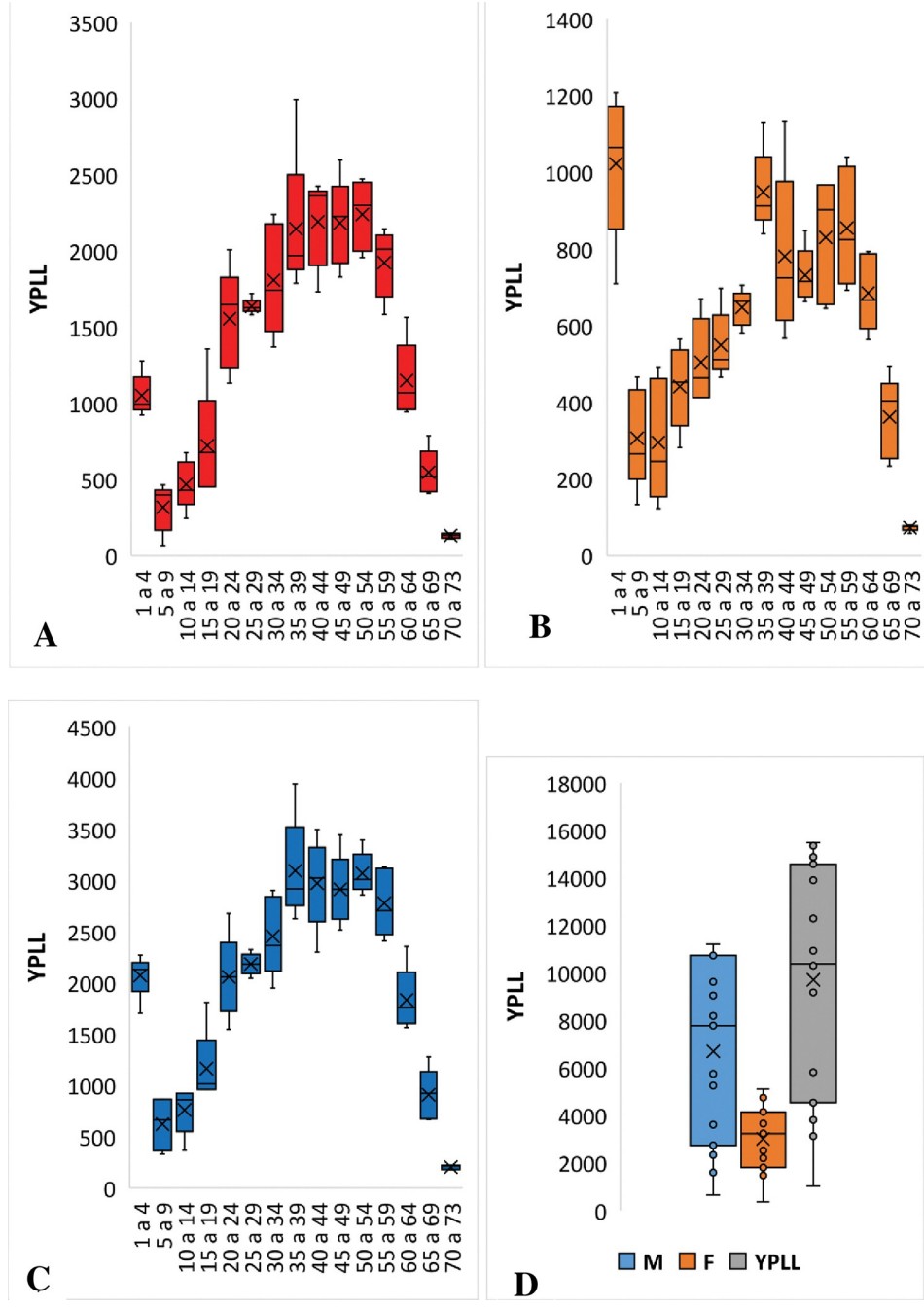

**Fig 1. Distribution of years of potential life lost by sex and age group, 2016 to 2020.** A is the distribution of YPLL for males, B is the distribution of YPLL for males and females, C and D represent the distribution of YPLL for both sexes.

system 15.9% (12,878). Injuries, poisoning and certain other consequences of external causes accounted for 11.2% (9,038), while neoplasms [tumors] and diseases of the respiratory system were responsible for 10.6% (8,583) and 8.1% (6,528) of total YPPLL, respectively Table 7.

Males accounted for 72.1% (58,403) of YPLL, with an average of 17.9 YPLL per death, compared to 27.9% (22,563) for females, with a mean of 16.5 YPLL per death. Overall, external causes of morbidity and mortality were responsible for 14,525 (17.9%) of YPPLL with an

**Table 6. Years of potential productive life lost by sex, 2016 to 2020.**

| Year | | Male | | | | Female | | | | Total | | |
|------|-----|--------|------------|------|-----|--------|------------|------|------|--------|------------|------|
| | n | YPPLL | YPPLL rate | Mean | n | YPPLL | YPPLL rate | Mean | n | YPPLL | YPPLL rate | Mean |
| 2016 | 628 | 12095.0 | 68.1 | 19.3 | 273 | 4622.5 | 27.1 | 16.9 | 901 | 16717.5 | 48.0 | 18.6 |
| 2017 | 579 | 9847.5 | 54.4 | 17.0 | 246 | 4195.0 | 24.2 | 17.1 | 825 | 14042.5 | 39.6 | 17.0 |
| 2018 | 690 | 12565.0 | 68.1 | 18.2 | 299 | 4762.5 | 27.1 | 15.9 | 989 | 17327.5 | 48.1 | 17.5 |
| 2019 | 639 | 11502.5 | 61.2 | 18.0 | 249 | 4142.5 | 23.3 | 16.6 | 888 | 15645.0 | 42.8 | 17.6 |
| 2020 | 731 | 12392.5 | 64.8 | 17.0 | 300 | 4840.0 | 26.9 | 16.1 | 1031 | 17232.5 | 46.5 | 16.7 |

average of 27.9 YPPLL per death. Results demonstrated that the major contributors to YPPLL for males were external causes of morbidity and mortality (22.0%; 12,843) with a mean of 28.1 per death. As for the female population, certain infectious and parasitic diseases were the principal contributors (24.6%; 5,560), with an average of 18.7 YPPLL per death Table 8.

The highest proportion of losses occurred in the 35–39 and 40–44 age groups and represented 27.3% and 28.2% of the total YPPLL, in the male and female population, respectively. The highest YPPLL recorded was in the 35 to 39 age range and corresponded to 2255 YPPLL while the lowest YPPLL was 205 in the 60 to 64 years age group, for males. For females, the 35 to 39 age range registered the highest number of YPPLL corresponding to 852 and the minimum record of YPPLL, 123, was in the age group between 60 and 64 years (Fig 2).

## Economic impact of premature death

Premature mortality in the period from 2016 to 2020, resulted in 98,659,153.23 USD of costs of productivity lost (CPL) with a mean of 21,290.28 USD per death. The CPL varied from

**Table 7. Evolution of years of potential productive life lost, 2016 to 2020.**

| YPPLL by cause of death | | | | | |
|---|---|---|---|---|---|
| Cause of death | Reference year | | | | |
| | 2016 | 2017 | 2018 | 2019 | 2020 |
| External causes of morbidity and mortality | 3723 | 2598 | 2845 | 2525 | 2835 |
| Diseases of the genitourinary system | 123 | 115 | 135 | 43 | 133 |
| Diseases of the circulatory system | 2548 | 2370 | 2580 | 2493 | 2888 |
| Diseases of the digestive system | 985 | 908 | 773 | 890 | 850 |
| Diseases of the respiratory system | 1030 | 1003 | 1675 | 1280 | 1540 |
| Mental and behavioral disorders | 550 | 595 | 540 | 328 | 603 |
| Endocrine, nutritional and metabolic diseases | 330 | 233 | 415 | 560 | 255 |
| Diseases of the nervous system | 543 | 265 | 425 | 345 | 385 |
| Pregnancy, childbirth and puerperium | 65 | 158 | 120 | 353 | 228 |
| Certain infectious and parasitic diseases | 2458 | 2233 | 3343 | 2355 | 3195 |
| Injury, poisoning and certain other consequences of external causes | 2158 | 1558 | 1915 | 1863 | 1545 |
| Congenital malformations, deformations and chromosomal abnormalities | 0 | 48 | 0 | 0 | 0 |
| Diseases of the musculoskeletal system and connective tissue | 0 | 43 | 55 | 0 | 3 |
| Diseases of the skin and subcutaneous tissue | 13 | 25 | 0 | 0 | 13 |
| Diseases of the blood and blood-forming organs and certain disorders involving the immune mechanism | 0 | 33 | 58 | 75 | 33 |
| Symptoms, signs and abnormal clinical and laboratory findings, not elsewhere classified | 478 | 425 | 768 | 773 | 750 |
| Neoplasms [tumors] | 1718 | 1438 | 1683 | 1765 | 1980 |
| **Total** | **16718** | **14043** | **17328** | **15645** | **17233** |

**Table 8. Years of potential productive life lost by cause of death, 2016 to 2020.**

| Cause of death | Male | | | Female | | | Total | | |
|---|---|---|---|---|---|---|---|---|---|
| | YPPLL | % | Mean | YPPLL | % | Mean | YPPLL | % | Mean |
| External causes of morbidity and mortality | 12842.5 | 22.0 | 28.1 | 1682.5 | 7.5 | 26.7 | 14525 | 17.9 | 27.9 |
| Diseases of the genitourinary system | 397.5 | 0.7 | 12.8 | 150.0 | 0.7 | 9.4 | 547.5 | 0.7 | 11.6 |
| Diseases of the circulatory system | 8582.5 | 14.7 | 13.6 | 4295.0 | 19.0 | 12.9 | 12877.5 | 15.9 | 13.3 |
| Diseases of the digestive system | 3095.0 | 5.3 | 16.1 | 1310.0 | 5.8 | 18.7 | 4405 | 5.4 | 16.8 |
| Diseases of the respiratory system | 4747.5 | 8.1 | 15.4 | 1780.0 | 7.9 | 16.8 | 6527.5 | 8.1 | 15.7 |
| Mental and behavioral disorders | 2377.5 | 4.1 | 17.4 | 237.5 | 1.1 | 18.3 | 2615 | 3.2 | 17.4 |
| Endocrine, nutritional and metabolic diseases | 1285.0 | 2.2 | 16.5 | 507.5 | 2.2 | 13.0 | 1792.5 | 2.2 | 15.3 |
| Diseases of the nervous system | 1460.0 | 2.5 | 20.9 | 502.5 | 2.2 | 26.4 | 1962.5 | 2.4 | 22.1 |
| Pregnancy, childbirth and puerperium | 0.0 | 0.0 | 0.0 | 922.5 | 4.1 | 31.8 | 922.5 | 1.1 | 31.8 |
| Certain infectious and parasitic diseases | 8022.5 | 13.7 | 17.6 | 5560.0 | 24.6 | 18.7 | 13582.5 | 16.8 | 18.0 |
| Injury, poisoning and certain other consequences of external causes | 8135.0 | 13.9 | 22.7 | 902.5 | 4.0 | 21.0 | 9037.5 | 11.2 | 22.5 |
| Congenital malformations, deformations and chromosomal abnormalities | 0.0 | 0.0 | 0.0 | 47.5 | 0.2 | 47.5 | 47.5 | 0.1 | 47.5 |
| Diseases of the musculoskeletal system and connective tissue | 57.5 | 0.1 | 19.2 | 42.5 | 0.2 | 42.5 | 100 | 0.1 | 25.0 |
| Diseases of the skin and subcutaneous tissue | 27.5 | 0.0 | 9.2 | 22.5 | 0.1 | 22.5 | 50 | 0.1 | 12.5 |
| Diseases of the blood and blood-forming organs and certain disorders involving the immune mechanism | 97.5 | 0.2 | 19.5 | 100.0 | 0.4 | 16.7 | 197.5 | 0.2 | 18.0 |
| Symptoms, signs and abnormal clinical and laboratory findings, not elsewhere classified | 2340.0 | 4.0 | 20.2 | 852.5 | 3.8 | 17.4 | 3192.5 | 3.9 | 19.3 |
| Neoplasms [tumors] | 4935.0 | 8.4 | 11.8 | 3647.5 | 16.2 | 13.1 | 8582.5 | 10.6 | 12.3 |
| **Total** | **58402.5** | **100** | **17.9** | **22562.5** | **100** | **16.5** | **80965.0** | **100** | **17.5** |

17,285,109.09 USD to 21,463,604.36 USD, while the leading contributors of CPL were circulatory system diseases (18,843,260.42 USD) followed by external causes of morbidity and mortality (12,316,441.51 USD), certain infectious and parasitic diseases (16,633,842.70 USD), and neoplasms (13,580,684.86USD). These causes combined accounted for 62.2% of the total CPL Tables 9 and 10.

Males accounted for 71.8% (70,818,833.31 USD) with a mean of 21,677.02 USD per death while females were responsible for 28.2% (27,840,319.93 USD) of the total CPL, and had an average of 20,366.00 USD per death. In general, the highest CPLs were recorded in the age groups between 35 and 59 years (73.7%; 72,715,770.69 USD) Table 10. Analysis showed that Pregnancy, childbirth and puerperium had the highest mean CPL per death of 26,116.26 USD, followed by the "external causes of morbidity and mortality" group, with an average CPL of 23,685.46 USD and Mental and behavioral disorders with a mean of 23,370.69 USD per death Table 10.

The sensitivity analysis performed, indicated total CPL of 134,724,671.62 USD (mean = 29 073.08 USD /death) at a discount rate of 3% and 74,517,897.02 USD (mean = 16,080.69 USD /death) at 6%. Diseases of the circulatory system had the highest CPL observed at a rate of 3%, and accounted for 24,297,098.13 USD or 18.0% of total CPL, followed by certain infectious and parasitic diseases with 22,809,540.03 USD (16.9%). At a rate of 6%, the same causes had the top two highest CPL Table 11.

## Discussion of results

The results of the present study point to a significant difference between the sexes, in terms of productivity losses. Males accounted for 71.8% of the total CPL. This result was higher than the results obtained in Tanzania and Iran, where the costs attributed to men were around 58% and 67%, respectively [4, 18]. However, in these two studies, authors evaluated the main causes

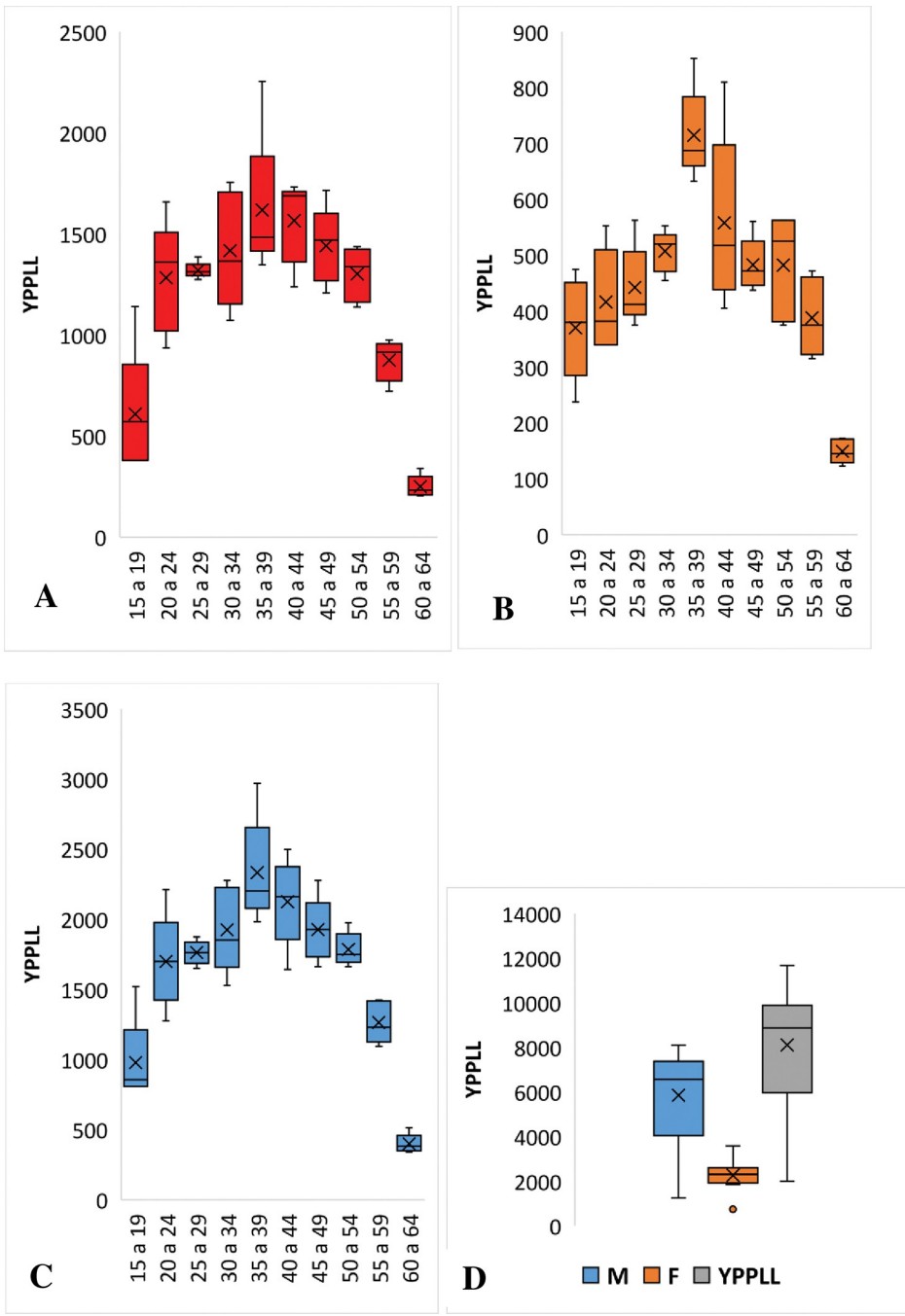

**Fig 2. Distribution of years of potential productive life lost by sex and age group, 2016 to 2020.** A is the distribution of YPPLL for males, B represents the distribution of YPPLL for males and females, C and D–represent the distribution of YPPLL for both sexes.

of death, while in the present study, all causes of premature death were taken into account [4, 18]. Furthermore, Łyszczarz [19] reported costs of lost productivity attributed to males of between 64.7% and 81.2% of the total costs, in Europe, while Díaz-Jiménez et al. [20] reported CPL for males three times higher than for females. The predominance of males shown in the current analysis reflected the burden of premature mortality in the sample population. In

**Table 9. Cost of productivity lost (in USD) by cause of death, 2016 to 2020.**

| Cause of death | 2016 | 2017 | 2018 | 2019 | 2020 |
|---|---|---|---|---|---|
| External causes of morbidity and mortality | 3073925.23 | 2183315.84 | 2415893.49 | 2194716.16 | 2448590.80 |
| Diseases of the genitourinary system | 167764.42 | 229093.17 | 256334.60 | 75154.97 | 168204.78 |
| Diseases of the circulatory system | 3564620.47 | 3539694.15 | 3808255.25 | 3561628.71 | 4369061.84 |
| Diseases of the digestive system | 1270545.30 | 1168387.24 | 1024750.91 | 1160451.01 | 1090518.36 |
| Diseases of the respiratory system | 1258969.32 | 1423126.38 | 2138614.40 | 1655213.04 | 1989788.77 |
| Mental and behavioral disorders | 736558.73 | 765967.70 | 764596.89 | 445149.45 | 793331.30 |
| Endocrine, nutritional and metabolic diseases | 522626.14 | 357363.09 | 592988.95 | 614479.74 | 410955.22 |
| Diseases of the nervous system | 598385.81 | 269465.21 | 387155.30 | 379429.36 | 414897.85 |
| Pregnancy, childbirth and puerperium | 51912.59 | 131210.84 | 107709.77 | 286243.07 | 180295.36 |
| Certain infectious and parasitic diseases | 2909533.92 | 2760117.89 | 3961300.28 | 3107916.68 | 3894973.93 |
| Injury, poisoning and certain other consequences of external causes | 2114697.85 | 1474031.68 | 1999671.55 | 2010407.09 | 1665704.75 |
| Congenital malformations, deformations and chromosomal abnormalities | 0.00 | 19796.71 | 0.00 | 0.00 | 0.00 |
| Diseases of the musculoskeletal system and connective tissue | 0.00 | 22073.43 | 51587.90 | 0.00 | 7552.18 |
| Diseases of the skin and subcutaneous tissue | 24315.30 | 35735.31 | 0.00 | 0.00 | 24315.30 |
| Diseases of the blood and blood-forming organs and certain disorders involving the immune mechanism | 0.00 | 26213.63 | 107040.73 | 47437.97 | 66811.36 |
| Symptoms, signs and abnormal clinical and laboratory findings, not elsewhere classified | 626656.88 | 478989.03 | 788430.13 | 847065.51 | 958750.44 |
| Neoplasms [tumors] | 2726208.00 | 2400527.81 | 2784972.24 | 2689124.69 | 2979852.12 |
| **Total** | **19646719.95** | **17285109.09** | **21189302.39** | **19074417.45** | **21463604.36** |

other economic studies conducted by Menzin et al. and Carter et al. [17, 21], the notable burden of lost productivity attributed to males was associated with greater participation of males in economic activities and disparities in income between the sexes.

The study showed that the estimated 80,965 YPPLL calculated resulted in a total loss of 98,659,153.23 USD, an annual average loss of 19,731, 830.65 USD and an average 21,290.28 USD per death. Total estimated CPL represented 5.3% of GDP (1,853,485,384.00 USD) in 2017 [14]. In comparison, a study by Łyszczarz [19] showed potential losses between 0.6% and 3.2% of GDP in the evaluated European countries. While Díaz-Jiménez et al. [20] reported values between 1.6 and 3.0% and Rumisha et al. [4] reported values equivalent to 0.3% of GDP.

In this study, the mean CPL per death was 21,290.28 USD, a value higher than that reported by Rumisha et al. [4], and lower than the mean value per death reported by Najafi et al. [18]. In the current analysis, the group "pregnancy, childbirth and puerperium" represented the highest CPL average per death (26,116.26 USD) which corresponded to 0.8% of the estimated total CPL. In comparison, this group accounted for a total loss of productivity of less than 0.1% in an Australian study [21] and 1% of lost productivity in the African region [22].

Mortality data analyzed showed that the main drivers of CPL were injuries, external causes of morbidity and mortality (12.5%) and injury, poisoning and certain other consequences of external causes (9.4%) combined represented 21.9% of the total CPL. Diseases of the circulatory system accounted for (19.1%) of CPL followed by certain infectious and parasitic diseases 16.7% and neoplasms 13.8%. When comparing with other studies evaluating costs of productivity loss from all causes of premature death, Carter et al. [21] reported neoplasms (30%) as the biggest driver of lost productivity, followed by cardiovascular diseases (19%), unintentional injuries (15%) and intentional injuries (13%). In their study, Rumisha et al. [4] reported the predominance of communicable diseases, followed by neoplasms and injuries among the main contributors to CPL. While Díaz-Jiménez et al. [20] showed that injuries were responsible for 61% of lost productivity in their study.

**Table 10. Cost of productivity lost by sex, age group, and cause of death, 2016 to 2020.**

| Variables | | Male | | | Female | | | Total | | |
|---|---|---|---|---|---|---|---|---|---|---|
| | | CPL | % | Mean | CPL | % | Mean | CPL | % | Mean |
| **Age group** | 15 to 19 | 1266989.31 | 1.8 | 19796.71 | 772071.61 | 2.8 | 19796.71 | 2039060.92 | 2.1 | 19796.71 |
| | 20 to 24 | 3333087.37 | 4.7 | 22073.43 | 1081597.89 | 3.9 | 22073.43 | 4414685.25 | 4.5 | 22073.43 |
| | 25 to 29 | 4271753.73 | 6.0 | 24271.33 | 1432008.35 | 5.1 | 24271.33 | 5703762.09 | 5.8 | 24271.33 |
| | 30 to 34 | 5714570.29 | 8.1 | 26213.63 | 2044662.76 | 7.3 | 26213.63 | 7759233.06 | 7.9 | 26213.63 |
| | 35 to 39 | 8126531.09 | 11.5 | 27641.26 | 3593364.09 | 12.9 | 27641.26 | 11719895.17 | 11.9 | 27641.26 |
| | 40 to 44 | 9807727.50 | 13.8 | 28183.13 | 3494707.50 | 12.6 | 28183.13 | 13302435.01 | 13.5 | 28183.13 |
| | 45 to 49 | 11254425.90 | 15.9 | 27316.57 | 3769686.34 | 13.5 | 27316.57 | 15024112.24 | 15.2 | 27316.57 |
| | 50 to 54 | 12668268.80 | 17.9 | 24315.30 | 4692851.97 | 16.9 | 24315.30 | 17361120.77 | 17.6 | 24315.30 |
| | 55 to 59 | 10599388.33 | 15.0 | 18180.77 | 4708819.17 | 16.9 | 18180.77 | 15308207.51 | 15.5 | 18180.77 |
| | 60 to 64 | 3776090.99 | 5.3 | 7552.18 | 2250550.23 | 8.1 | 7552.18 | 6026641.22 | 6.1 | 7552.18 |
| | **Total** | **70818833.31** | **100** | **21677.02** | **27840319.93** | **100** | **20366.00** | **98659153.23** | **100** | **21290.28** |
| **Cause of death** | External causes of morbidity and mortality | 10921623.04 | 15.4 | 23898.52 | 1394818.46 | 5.0 | 22139.98 | 12316441.51 | 12.5 | 23685.46 |
| | Diseases of the genitourinary system | 629666.70 | 0.9 | 20311.83 | 266885.24 | 1.0 | 16680.33 | 896551.94 | 0.9 | 19075.57 |
| | Diseases of the circulatory system | 12494202.84 | 17.6 | 19738.08 | 6349057.58 | 22.8 | 19009.15 | 18843260.42 | 19.1 | 19486.31 |
| | Diseases of the digestive system | 4223457.79 | 6.0 | 21997.18 | 1491195.04 | 5.4 | 21302.79 | 5714652.83 | 5.8 | 21811.65 |
| | Diseases of the respiratory system | 6361743.65 | 9.0 | 20588.17 | 2103968.27 | 7.6 | 19848.76 | 8465711.91 | 8.6 | 20399.31 |
| | Mental and behavioral disorders | 3203189.21 | 4.5 | 23380.94 | 302414.87 | 1.1 | 23262.68 | 3505604.08 | 3.6 | 23370.69 |
| | Endocrine, nutritional and metabolic diseases | 1769101.58 | 2.5 | 22680.79 | 729311.56 | 2.6 | 18700.30 | 2498413.15 | 2.5 | 21353.96 |
| | Diseases of the nervous system | 1618362.17 | 2.3 | 23119.46 | 430971.36 | 1.5 | 22682.70 | 2049333.53 | 2.1 | 23026.22 |
| | Pregnancy, childbirth and puerperium | 0.00 | 0.0 | 0.00 | 757371.63 | 2.7 | 26116.26 | 757371.63 | 0.8 | 26116.26 |
| | Certain infectious and parasitic diseases | 10154650.54 | 14.3 | 22220.24 | 6479192.15 | 23.3 | 21742.26 | 16633842.70 | 16.9 | 22031.58 |
| | Injury, poisoning and certain other consequences of external causes | 8370649.76 | 11.8 | 23381.70 | 893863.15 | 3.2 | 20787.52 | 9264512.91 | 9.4 | 23103.52 |
| | Congenital malformations, deformations and chromosomal abnormalities | 0.00 | 0.0 | 0.00 | 19796.71 | 0.1 | 19796.71 | 19796.71 | 0.0 | 19796.71 |
| | Diseases of the musculoskeletal system and connective tissue | 59140.08 | 0.1 | 19713.36 | 22073.43 | 0.1 | 22073.43 | 81213.50 | 0.1 | 20303.38 |
| | Diseases of the skin and subcutaneous tissue | 56182.77 | 0.1 | 18727.59 | 28183.13 | 0.1 | 28183.13 | 84365.90 | 0.1 | 21091.47 |
| | Diseases of the blood and blood-forming organs and certain disorders involving the immune mechanism | 108656.67 | 0.2 | 21731.33 | 138847.02 | 0.5 | 23141.17 | 247503.68 | 0.3 | 22500.33 |
| | Symptoms, signs and abnormal clinical and laboratory findings, not elsewhere classified | 2706137.57 | 3.8 | 23328.77 | 993754.41 | 3.6 | 20280.70 | 3699891.98 | 3.8 | 22423.59 |
| | Neoplasms [tumors] | 8142068.95 | 11.5 | 19478.63 | 5438615.92 | 19.5 | 19493.25 | 13580684.86 | 13.8 | 19484.48 |
| | **Total** | **70818833.31** | **100** | **21677.02** | **27840319.93** | **100** | **20366.00** | **98659153.23** | **100** | **21290.28** |

The 921 deaths verified related to injuries and external causes in this study, occurred mostly in the young population and contributed to about 23,562.5 YPPLL, which translated in 21,580,954.42 USD (21.9% of total CPL). For this cause of death, 78% of YPPLL were attributed to the age groups between 20 to 44 years. Showing that death due to injuries and external causes occurred in younger age groups. The results of this study corroborate findings by Delgado and Carter et al. [7, 21] that demonstrated that death due to injuries often occur in the younger population. In comparison, 967 deaths due to diseases of the circulatory system resulted in 12,877.5 YPPLL approximately 50% less than YPPLL due to injuries and external causes. Diseases of the circulatory system corresponded to CPL of 18,843,260.42 USD (19.1% of total CPL), and these deaths occurred mainly in the older age groups, the age group 40 to 59 years accounted for 59.4% of YPPLL due to this cause.

**Table 11. Lost productivity cost using GDP per capita at 3% and 6% discount rate, 2016 to 2020.**

| Cause of death | Present value (Total) | | | | | |
|---|---|---|---|---|---|---|
| | 3% | % | Mean | 6% | % | Mean |
| External causes of morbidity and mortality | 19015041.77 | 14.1 | 36567.39 | 8250513.82 | 11.1 | 15866.37 |
| Diseases of the genitourinary system | 1122230.56 | 0.8 | 23877.25 | 729652.17 | 1.0 | 15524.51 |
| Diseases of the circulatory system | 24297098.13 | 18.0 | 25126.26 | 14990278.74 | 20.1 | 15501.84 |
| Diseases of the digestive system | 7694029.08 | 5.7 | 29366.52 | 4352584.69 | 5.8 | 16612.92 |
| Diseases of the respiratory system | 11334604.38 | 8.4 | 27312.30 | 6512545.57 | 8.7 | 15692.88 |
| Mental and behavioral disorders | 4703578.15 | 3.5 | 31357.19 | 2664327.91 | 3.6 | 17762.19 |
| Endocrine, nutritional and metabolic diseases | 3284790.53 | 2.4 | 28075.13 | 1946388.86 | 2.6 | 16635.80 |
| Diseases of the nervous system | 2939208.04 | 2.2 | 33024.81 | 1479318.09 | 2.0 | 16621.55 |
| Pregnancy, childbirth and puerperium | 1204568.78 | 0.9 | 41536.85 | 482418.45 | 0.6 | 16635.12 |
| Certain infectious and parasitic diseases | 22809540.03 | 16.9 | 30211.31 | 12456681.11 | 16.7 | 16498.92 |
| Injury, poisoning and certain other consequences of external causes | 13427871.31 | 10.0 | 33485.96 | 6598101.86 | 8.9 | 16454.12 |
| Congenital malformations, deformations and chromosomal abnormalities | 39626.72 | 0.0 | 39626.72 | 9988.34 | 0.0 | 9988.34 |
| Diseases of the musculoskeletal system and connective tissue | 126469.38 | 0.1 | 31617.35 | 54453.07 | 0.1 | 13613.27 |
| Diseases of the skin and subcutaneous tissue | 105874.72 | 0.1 | 26468.68 | 67948.23 | 0.1 | 16987.06 |
| Diseases of the blood and blood-forming organs and certain disorders involving the immune mechanism | 332274.62 | 0.2 | 30206.78 | 190714.50 | 0.3 | 17337.68 |
| Symptoms, signs and abnormal clinical and laboratory findings, not elsewhere classified | 5151758.37 | 3.8 | 31222.78 | 2737935.51 | 3.7 | 16593.55 |
| Neoplasms [tumors] | 17136107.06 | 12.7 | 24585.52 | 10994046.11 | 14.8 | 15773.38 |
| **Total** | **134724671.62** | **100** | **29073.08** | **74517897.02** | **100** | **16080.69** |

The results of the present study showed that the highest productivity loss was due to non-communicable diseases. For example, diseases of the circulatory system and neoplasms represented 32.9% of the total CPL. This finding is in line with the results of studies carried out in Iran [18] and in Australia [21], but differed from results in Tanzania [4] which reported a predominance of communicable diseases. However, communicable diseases continue to represent an important burden in the mortality profile of Cabo Verde. For instance, certain infectious and parasitic diseases accounted for 16.7% of CPL, showing that the country is still in an epidemiological transition [23]. Interestingly, this cause of death was the main driver of CPL in the female population, accounting for 23.3% of the total CPL in this group, while diseases of the circulatory system and neoplasms contributed with 22.8% and 19.5%, respectively. Evidence available on the economic burden of disease on the African Continent [22] indicated a near balance between non-communicable diseases (37%) and communicable diseases (36%) as the main drivers of productivity losses, and this represents a challenge for health systems that have traditionally focused on communicable diseases [24]. It also emphasizes the need for greater investment in health and demonstrates the need to implement robust public health policies that address these issues [22].

Sensitivity analysis was performed according to the guidelines for economic studies [25, 26]. The discount rates of 3% and 6% were applied, which resulted in present values of lost productivity of 134,724,671.62 USD and 74,517,897.02 USD, respectively. These values correspond to 7.3% and 4.0% of 2017 GDP [14].

There is a limited number of studies that assess productivity loss due to all causes of death, particularly on the African continent. In addition, most studies found analyzed costs due to specific causes [17, 27–33]. Furthermore, no studies were found that addressed the issue of productivity loss in small island developing countries, such as Cabo Verde. Thus, the present study will contribute to filling in the gap on knowledge on the subject at national and regional level.

This study had limitations that should be taken into account when interpreting the results. Firstly, it was assumed that each individual of working age would contribute equally to society if they did not die by the defined age limit using the Human Capital approach. Although this method is often criticized as it tends to overestimate the losses resulting from premature mortality [12, 17, 34], however, the method is widely used in economic studies, due to its relatively easy application. Moreover, there is still no consensus on the best methodology to assess the indirect costs of mortality [5, 12, 35]. Future economic studies evaluating the costs of premature death in Cabo Verde, could apply other methods described in literature [17, 19, 21].

Other limitations include that fact that other indirect productivity losses related to mortality were not considered in the analysis, such as intangible costs, among others, which could lead to an underestimation of the economic burden of the analyzed causes. Cost analysis was performed by group of causes. Therefore, the loss of productivity due to specific deaths was not demonstrated. This may constitute a topic for future investigations.

Finally, analysis determined that 4% of the calculated costs were attributed to the group of "symptoms, signs and abnormal findings of clinical and laboratory tests, not classified elsewhere", which highlights the need to further improve the classification of causes of death in the country's health information system.

## Conclusion

The study demonstrated the burden of premature mortality and associated costs in Cabo Verde from 2016 to 2020 and elucidated patterns of early mortality in the period analyzed. Injuries and external causes were the main causes premature mortality and the principal drivers of lost productivity in the study period, followed by diseases of the circulatory system, certain infectious and parasitic diseases and neoplasms. Non communicable diseases represented an important disease burden with regards to the costs of productivity lost in Cabo Verde.

The economic burden of premature mortality was substantial, corresponding to 98,659,153.23 USD during 2016–2020. The mean cost of productivity lost was 21,290.28 USD per death. Overall, the cost of productivity losses for the males was 2.5 times higher than the females.

The estimation of the burden of mortality and the cost of productivity lost for society and the national economy can be a valuable instrument for assessing the mortality profile of the country, thus complementing the measures traditionally used to demonstrate the burden of disease and support decision-making and allocation of resources. The implementation of policies aimed at reducing premature mortality, especially from preventable causes, may result in reduced costs of lost productivity and improvements in the well-being of the Cape-Verdean population.

## Supporting information

**S1 Appendix. Characterization of deaths from all causes, Cabo Verde, 2016 to 2020.**
(DOCX)

**S2 Appendix. Number of deaths by causes, Cape Verde, 2016 to 2020.**
(DOCX)

**S3 Appendix. Rates of years of potential life lost by municipality, Cabo Verde, 2016 to 2020.**
(DOCX)

**S4 Appendix. Rates of years of potential productive life lost by municipality, Cabo Verde, 2016 to 2020.**
(DOCX)

## Acknowledgments

The authors would like to thank the staff in charge of compiling mortality data in the Ministry of Health and the National Health Directorate for authorizing the use of data.

## Author Contributions

**Conceptualization:** Ngibo Mubeta Fernandes, Janilza Solange Gomes Silveira Silva, Edna Duarte Lopes.

**Data curation:** Domingos Veiga Varela, Janice de Jesus Xavier Soares.

**Formal analysis:** Ngibo Mubeta Fernandes, Janice de Jesus Xavier Soares.

**Investigation:** Ngibo Mubeta Fernandes, Janilza Solange Gomes Silveira Silva, Janice de Jesus Xavier Soares.

**Methodology:** Ngibo Mubeta Fernandes, Janice de Jesus Xavier Soares.

**Project administration:** Edna Duarte Lopes.

**Software:** Janice de Jesus Xavier Soares.

**Supervision:** Edna Duarte Lopes.

**Validation:** Ngibo Mubeta Fernandes, Janice de Jesus Xavier Soares.

**Visualization:** Ngibo Mubeta Fernandes, Janice de Jesus Xavier Soares.

**Writing – original draft:** Ngibo Mubeta Fernandes, Janilza Solange Gomes Silveira Silva, Domingos Veiga Varela, Janice de Jesus Xavier Soares.

**Writing – review & editing:** Ngibo Mubeta Fernandes, Janilza Solange Gomes Silveira Silva, Edna Duarte Lopes, Janice de Jesus Xavier Soares.

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
