## [Decision Letter · Decision Letter 0]

15 Feb 2023

PONE-D-22-31822The economic impact of premature mortality in Cabo Verde: 2016 - 2020PLOS ONE

Dear Dr. Fernandes,

Thank you for submitting your manuscript to PLOS ONE. After careful consideration, we feel that it has merit but does not fully meet PLOS ONE’s publication criteria as it currently stands. Therefore, we invite you to submit a revised version of the manuscript that addresses the points raised during the review process.

We look forward to receiving your revised manuscript.

Kind regards,

Bilal Sulaiman

Academic Editor

PLOS ONE

Journal Requirements:

Additional Editor Comments (if provided):

The reviewer has raised some observations that need to be addressed in your response.

Reviewers' comments:

Reviewer's Responses to Questions

**Comments to the Author**

1. Is the manuscript technically sound, and do the data support the conclusions?

Reviewer #1: Partly

2. Has the statistical analysis been performed appropriately and rigorously? 

Reviewer #1: Yes

3. Have the authors made all data underlying the findings in their manuscript fully available?

Reviewer #1: Yes

4. Is the manuscript presented in an intelligible fashion and written in standard English?

Reviewer #1: No

5. Review Comments to the Author

Reviewer #1: the paper in general was well written. I have some issues which I need the author to address. I felt the terms YPPL and CPL should be defined not just YPLL to enable readers understand the concept of the article. line 160 , table 1, age ranges used is not conventional, I advise re-analysing using decades of life. Why is the author lumping parasitic disease and certain infections?. Parasitic diseases are also infections.... what certain diseases does the author means in this study? The author should state what accounted for the remaining 18.3% of deaths among the study population on line 165. Table 2, line 170, causes of deaths should be arranged using the variable with highest frequency and then in a decreasing fashion. I have detected grammatical errors and incorrect use of tenses which should be addressed. Punctuation marks should be inserted appropriately for example on line 184, comma should be inserted after'. table'. I felt the conclusion was not concise and has not summarised the findings from the study as such, the main cause of premature mortality should be stated clearly.

6. PLOS authors have the option to publish the peer review history of their article (what does this mean?). If published, this will include your full peer review and any attached files.

Reviewer #1: No

---

## [Author Response · Author response to Decision Letter 0]

22 Mar 2023

Dear Editor,

We would like to thank the editor and reviewers for their comments on our manuscript-The economic impact of premature mortality in Cabo Verde: 2016 - 2020

Following are our responses to the reviewers’ comments.

.Reviewer #1: the paper in general was well written. I have some issues which I need the author to address. 

1. I felt the terms YPPL and CPL should be defined not just YPLL to enable readers understand the concept of the article. 

Ans. Definations of YPPLL and CPL have been provided

2. line 160, table 1, age ranges used is not conventional, I advise re-analysing using decades of life. 

Ans. Line 160 Table 1 have been corrected according to WHO standards.

3. Why is the author lumping parasitic disease and certain infections ? Parasitic diseases are also infections.... what certain diseases does the author means in this study? 

Ans.We used the description used in International Statistical Classification of Diseases and Related Health Problems, 10th Revision to classify the groups of causes of death. 

https://icd.who.int/browse10/2016/en

 Certain infectious and parasitic diseases refering to diseases with codes A00 to B99

4. The author should state what accounted for the remaining 18.3% of deaths among the study population on line 165. 

Ans. 18.3% was distributed among the rest of the causes of death in the table. We only mentioned the principal causes in our manuscript.

5. Table 2, line 170, causes of deaths should be arranged using the variable with highest frequency and then in a decreasing fashion. 

Ans. Table 2, line 170 has been corrected as suggested.

6. I have detected grammatical errors and incorrect use of tenses which should be addressed. Punctuation marks should be inserted appropriately for example on line 184, comma should be inserted after'. table'. 

Ans. Corrections were done as suggested. 

7. I felt the conclusion was not concise and has not summarised the findings from the study as such, the main cause of premature mortality should be stated clearly.

Ans. We revised the conclusion as suggested 

Once again, our thanks to the editor and reviewers 

Best regards

Ngibo Mubeta Fernandes

---

## [Editor Report · Decision Letter 1]

2 May 2023

The economic impact of premature mortality in Cabo Verde: 2016 - 2020

PONE-D-22-31822R1

Dear Dr. Fernandes,

We’re pleased to inform you that your manuscript has been judged scientifically suitable for publication and will be formally accepted for publication once it meets all outstanding technical requirements.

Kind regards,

Bilal Sulaiman

Academic Editor

PLOS ONE

Additional Editor Comments (optional):

All comments have been addressed appropriately.
---

## [Editor Report · Acceptance letter]

16 May 2023

PONE-D-22-31822R1 

The economic impact of premature mortality in Cabo Verde: 2016 - 2020 

Dear Dr. Fernandes:

I'm pleased to inform you that your manuscript has been deemed suitable for publication in PLOS ONE. Congratulations! Your manuscript is now with our production department. 

Kind regards, 

on behalf of

Dr. Bilal Sulaiman 

Academic Editor

PLOS ONE